# Factors Associated with Esthetic Outcomes of Flapless Immediate Placed and Loaded Implants in the Maxillary Incisor Region—Three-Year Results of a Prospective Case Series

**DOI:** 10.3390/jcm12072625

**Published:** 2023-03-31

**Authors:** Edith Groenendijk, Tristan Ariaan Staas, Ewald Maria Bronkhorst, Gerry Max Raghoebar, Gert Jacobus Meijer

**Affiliations:** 1Private Clinic for Oral Implantology & Reconstructive Dentistry, Implantologie Den Haag BV, Stadhouderslaan 12, 2517 HW The Hague, The Netherlands; 2Private Dental Clinic, Staas & Bergmans Zorg Voor uw Mond, Schubertsingel 32, 5216 XA Den Bosch, The Netherlands; 3Radboud Institute of Health Sciences, Department of Dentistry, Radboud University Medical Center, Geert Grooteplein Zuid 10, 6525 GA Nijmegen, The Netherlands; 4Department of Oral and Maxillofacial Surgery, University of Groningen, University Medical Center Groningen, Hanzeplein 1, 9713 GZ Groningen, The Netherlands; 5Department of Oral and Maxillofacial Surgery, Radboudumc, Geert Grooteplein Zuid 10, 6525 GA Nijmegen, The Netherlands

**Keywords:** immediate implants, esthetic outcome (PES/modPES/WES), flapless, soft-tissue recession, immediate loading, buccal gap, buccal bone defect, palatal implant position, bovine bone substitute

## Abstract

Flapless immediate implant placement and provisionalization (FIIPP) is often associated with an increased risk of buccal soft-tissue recession. This study aims to assess the 3-year esthetic outcome. In 100 consecutive patients, one maxillary incisor, with or without a pre-extraction buccal bone defect (≤5 mm), was replaced by an implant installed in a maximal palatal position (buccal gap ≥2 mm). The created gaps were filled with bovine bone substitute. Patient satisfaction (PS), pink esthetic scores (PES/modPES), and white esthetic score (WES) were calculated at different time points. A multilevel regression analysis (MRA) was performed to analyze which factors may be associated with the esthetics. After three years, PS scored 8.9 ± 0.84 on a scale of 10 (*n* = 83), and the soft-tissue esthetics were high (PES = 12.2; modPES = 8.5), as was the WES (8.2), showing no decrease from one year. Buccal bone defect size and smoking could not be associated with the soft-tissue outcome; however, implant location, gap size, and emergence profiles could. Performing FIIPP, the final crown (WES) scored highest when it was cemented, the soft tissue (PES/modPES) in central-incisor positions, and all (WES/PES/modPES) with concave emergence profiles.

## 1. Introduction

Single tooth replacement with a dental implant in the anterior maxillary region is extremely challenging. One subtle mistake in indication, treatment planning, the surgical or the restorative protocol and/or the performance of the clinician may lead to a catastrophic esthetic outcome. To diminish morbidity and optimize patient comfort, minimally invasive flapless immediate implant placement and provisionalization (FIIPP) should be the clinical protocol of first choice. Nevertheless, in the esthetic zone of the maxilla, FIIPP is considered complex, with a higher risk of buccal soft-tissue recession and a compromised esthetic outcome compared to early or delayed implant placement [1]. For this reason, it is generally recommended to perform immediate implant placement only in sites with an intact socket, a thick buccal bone crest (>1 mm), and a firm gingival phenotype [2,3,4]. Unfortunately, such ideal conditions are scarce; only 13% of patients have a thick buccal bone crest [5].

To achieve acceptable soft-tissue esthetics, healing of the socket by delaying implant placement is suggested [6]. However, vascularization of the buccal crest is removed by tooth extraction and the disappearance of the periodontal ligament, leading to the destruction of the thin cortical plate. Therefore, post-extraction alveolar remodeling is inevitable; human re-entry studies showed horizontal and vertical bone loss of 29–63% and 11–22%, respectively, six months following tooth extraction [7]. In thin buccal crests (<1 mm), an average of 7.5 mm mid-buccal vertical bone loss was reported within the first 8 weeks post extraction [8].

There is evidence that three-dimensional implant positioning influences the bony dimensions around implants and hence the esthetic outcome of the soft tissue (ST). A buccal crest thickness of ≥2 mm in front of the implants seems to be a favorable basis for an ideal and harmonic soft-tissue condition that remains stable over a long period [9]. Most soft-tissue recessions following implant treatment can be explained by surgical protocols advising a more buccal implant position [10,11,12,13,14]. By contrast, the presented clinical protocol proposes a maximal palatal implant position to create a gap of at least 2 mm from the buccal crest. Other factors may also influence the esthetic outcome, for instance preoperative esthetics, smoking, buccal bone defects, implant location within the arch, a cemented or screw-retained crown, a convex versus concave emergence profile, and the occurrence of complications.

Due to the heterogeneity of the published research, such as in the procedures followed and the materials used, there is no consensus as to which protocol is best for replacing a single tooth in the esthetic zone. Furthermore, different esthetic indices are used in the various studies. To describe soft-tissue esthetics, Fürhauser et al. (2005) defined the pink esthetic score (PES) [15], an index of 7 parameters scored 0, 1 or 2, resulting in a PES index of 0 to 14. In 2009, the modified-PES (modPES) [16] merged PES-5 (alveolar process deficiency) with PES-6 (ST color) and PES-7 (ST texture), thereby reducing the number of parameters to five and, resulting in an index score of 0 to 10. The modPES-version to evaluate ST esthetics was used in three studies on early implant placement (EIP) [17,18,19], one on delayed implant placement (DIP) [20], four on immediate implant placement (IIP) [21,22,23,24], and in one randomized clinical trial on IIP versus DIP [25]. Meanwhile, most IIP studies [26,27,28,29,30,31,32,33,34,35,36,37,38,39,40] used the original PES index.

Priority should be given to those treatments that reach an optimal result in a minimally invasive way and with the shortest treatment time. The one-year interim results of the present study have been previously reported [36]. They demonstrated high esthetic outcomes: mean WES improved from a score of 4.5 to 8.2 (*p* = 0.00), and PES improved from 9.9 to 12.1 (*p* = 0.00). The aim of this prospective clinical trial is to assess 3-year postoperative patient satisfaction, esthetic outcomes and identify factors to FIIPP outcomes. The hypothesis is that the one-year esthetic results remain stable at three years.

## 2. Materials and Methods

In this multicenter prospective cohort study, one hundred consecutive patients were enrolled after providing written informed consent for participation in this study and publication of their data. Inclusion criteria consisted of one failing single maxillary incisor between two healthy neighboring teeth (1), in the presence of sufficient occlusal support (2), and adequate vertical bone height at the palato-apical part of the socket (≥ 5 mm) to support primary implant stability (3). Based on the preoperative CBCT, both intact sockets and sockets with a periapical bone defect and/or buccal bone defect of ≤5 mm with the cement–enamel junction as reference were allowed (4). Patients with periodontitis, bruxism, smoking more than 10 units per day, pregnancy, drug or alcohol abuse, or a general health issue which elicits negative bone reactions were excluded.

Patient enrollment, FIIPP and evaluations were performed in the years 2014 to 2017, and data collection and analyses took place till 2022. Ethical approval was acquired from the Ethics Committee of the Radboud University Medical Center Nijmegen (2014/157). This research was registered in the Dutch Trial Register (NTR) on 20 October 2015 (NTR5583/NL4170). Patients were treated in six treatment centers. In two centers, an oral maxillofacial surgeon performed the surgery, and a prosthodontist performed the restorative procedure. In the remaining four centers, dentists completed both the surgical and the restorative procedures.

### 2.1. Clinical Procedure

The clinical procedure was described in detail in earlier publications [32,36,39]. Briefly, patients were instructed to take 2 g amoxicillin, one hour preoperatively, followed by 500 mg amoxicillin every eight hours for five days, and to rinse with 0.12% chlorhexidine solution twice daily for 14 days after. Prior to the procedure, a low-dose, small-field-of-view (FOV) CBCT was acquired. After atraumatic tooth removal, the socket was thoroughly cleaned by curettage. Without raising a flap, the first osteotomy was made using a sharp pointed precision drill into the palatal wall (approximately 2 mm coronally from half of the root length) when using the class 1 alveolar process (when the root was positioned buccally) [41]. When using the class 2 (when the root was positioned in the middle), class 3 (when the root was positioned to the palatal) or class 4 (when the root was filling the alveolar process) process, the osteotomy was performed in the long axis of the apex at the palatal side of the socket. A full-length osteotomy was made using a 2 mm twist drill, followed by larger-diameter twist drills according to the manufacturer’s guidelines (Nobel Biocare AB™, Karlskoga, Sweden). In advance of implant installation, the last used drill was repositioned into the implant bed, and the socket was filled with a bone substitute (Bio-Oss™ 0.25–1 mm, Geistlich Pharma AG, Wolhusen, Switzerland). Subsequently, the drill was carefully removed, and a variable-thread tapered implant with an internal conical connection (NobelActive™, Nobel Biocare AB) was positioned 3–4 mm sub-gingivally, with the mid-buccal gingival margin of the contralateral tooth as reference. A temporary non-loaded screw-retained crown was fabricated and installed in the same session. Check-up took place one to two weeks postoperatively. Between three and nine months later, the final crown was installed. Effort was made to individualize abutments in an optimal slender emerging profile (Procera™, Nobel Biocare AB, Mahwah, USA).

### 2.2. Clinical and Radiological Measurements

The buccal bone crest distance of the “the top of the mid-buccal gingiva to the bony crest” was measured directly post extraction to the nearest mm using a periodontal probe (Hu-Friedy, Frankfurt, Germany). Bone defects were defined as the distance of the “top of the mid-buccal gingiva to the bony crest minus 3 mm.

Before and after the procedure (T0, T1), after final crown placement (T2), and after one (T3) and three years (T4), periodontal health was evaluated according to the Dutch Periodontal Screening Index (DPSI) [42]. The highest DPSI score was detected and recorded for each sextant using a mirror and a periodontal probe (Hu-Friedy, Frankfurt, Germany). The highest score of the six sextants determines the DPSI index, which reflects the periodontal condition.

The emergence profile of the final crown was evaluated on CBCT and defined as “convex” when a straight or round profile was present, or “concave” when the profile was slender.

Gap size (distance between the inner buccal bone crest and the mid-buccal aspect of the implant seat) was measured (mm) on the postoperative CBCT (T1), using On Demand software version 1.0 (Cybermed Inc., Seoul, Korea) (Figure 1).

### 2.3. Esthetic Measurements

Both the implant and the contralateral tooth were photographed following a standardized protocol [43] at the following time points: preoperatively (T0), 7–14 days after (T1), after final crown placement (T2), and at one-year (T3), and 3-year follow up from implant placement (T4). At each time point, two light photographs were taken: one perpendicular to the mid-buccal axis of the tooth arch, and one perpendicular to the implant site. Light photographs were digitally stored in a raw format (PowerPoint, Microsoft, Redmond, Washington, USA). After a calibration session, two independent examiners, who were not involved in the patient treatment, evaluated the clinical photographs.

To allow optimal comparison with other studies, both the original PES (scale 0–14) [15], and the modified PES (scale 0–10) [16] were reported. Additionally, the white esthetic score (WES) [14] with a scale of 0–10 was assessed.

Some authors [22,24] proposed an esthetic soft-tissue classification based on the combined PES values such as “excellent” (PES = 12–14), “acceptable” (PES = 8–11), and “inadequate” (PES < 8).

To provide an “overall esthetic score”, WES and PES outcomes can be combined [26,29] and classified as “excellent” (PES ≥ 12 and WES ≥ 9) and “inadequate” (PES < 8 and/or WES < 6). To improve balance, two extra scores were added [36]; “good” (PES ≥ 10 and WES ≥ 8), and “acceptable” (PES ≥ 8 and WES ≥ 6).

### 2.4. Patient Satisfaction

To evaluate patients’ opinion concerning the surgical and restorative treatment and esthetic outcome, a validated patient satisfaction (PS) questionnaire [44] was used (Table 1). At the time points T1 to T4, nine individual questions were scored with values between 1 and 5. Questions 1 to 4 (Q1–Q4) inventoried the surgical procedure, and questions 5 to 8 (Q5–Q8) covered the esthetic outcome. For clarity, the total score (Q9) was transferred to a scale of 1–10.

### 2.5. Statistical Analysis

The mean, standard deviation (SD), and range were calculated at all time points for PS, PES, modPES, and WES. To evaluate the correlation between PS and PES, and between PS and WES, the two-tailed Pearson correlation was calculated.

The effect of different factors on WES, PES, and modPES was analyzed by means of a multilevel regression analysis (MRA); preoperative esthetics, smoking (0 = non-smoker/1 = smoking < 10 units a day), size of post-extraction mid-buccal bone defect (mm), implant location (1 = central/2 = lateral incisor site), gap size (mm), final crown (1 = screw retained/2 = cemented), emergence profile (1 = concave/2 = convex), complications (0 = no, 1 = yes), and time. In the MRA, the scores for PES, modPES and WES for all time points (T0–T4) were used. The measurement at T0 is an independent variable, and therefore a given, and the scores at the other times are outcomes. The modelling of ‘time’ in the MRA was performed by incrementally increasing the complexity, until no improvement of the model was seen. The steps were: no role of time, stepwise change over time, linear change, quadratic change, and cubic change. In all the options, time was treated as a discrete variable with values of 1, 2, 3, or 4. The MRA was accomplished using library lme4 (1.1–21) in R (version 3.6.3). Statistical significance was defined as *p* ≤ 0.05.

## 3. Results

The 3-year follow-up analysis included 83 patients (45 females and 38 males; mean age 46; range 17–80 years). In total, three patients were excluded due to clinical reasons; in one patient, a new trauma occurred, and the lost implant was replaced by a new implant. In the other two patients, a neighboring tooth was replaced by an implant. An additional 14 patients were lost due to relocation or refusal to attend follow-up visits. Of the remaining population, 12 patients (14%) smoked less than 10 cigarettes a day. In total, 54 central and 29 lateral incisors were replaced. Of these, 34 final crowns were screw retained and 49 cemented. In total, 55 crowns showed a concave and 28 a convex emergence profile.

### 3.1. Implant Survival and Complications

All implants were in function at three years, yielding an implant survival rate of 100%. The treatment centers of the patients who were lost to follow up were approached for information about the patients’ status; no implant loss was reported.

Two restorative complications were reported: one temporary crown loosening and one zirconium abutment fracture.

Biological complications were ascribed to treatment for breast and prostate cancer between T3 and T4. At the three-year evaluation, nine patients showed peri-implant mucositis (bleeding on probing; no bone loss around the implant seat).

### 3.2. General Periodontal Health

At T0, all patients showed a healthy periodontium (DPSI A). After three years (T4), in two patients, at least one pocket of 4–5 mm in the natural dentition with bleeding after probing without observable recession(s) above the deepened pocket(s) (DPSI B) was found, and the remaining 81 patients scored a healthy periodontium (DPSI A).

### 3.3. Buccal Bone Defects

The distribution of the size of post-extraction mid-buccal bone defects and related soft-tissue esthetic outcome (T4-PES) is shown in Table 2. Despite an atraumatic approach, eleven buccal crests were damaged or lost after extraction and/or thorough cleaning of the socket walls, resulting in buccal bone defects larger than 5 mm.

### 3.4. Bone Gap Measurements

Mean gap size was 2.7 ± 0.9 mm (range 1.3–6.4 mm).

### 3.5. Patient Satisfaction

Mean overall patient satisfaction at the different time points T1–T4 is shown in Figure 2. After immediate implant placement and temporary restoration (T1), PS was 8.5 (±0.98), and increased after inserting the final crown (T2) to 8.8 (±0.8), and one year postoperative (T3) to 9.0 (±0.8). Thereafter, patients remained consistent in their opinion (T4: 8.9 ± 0.8).

After three years, surgical treatment duration was experienced as moderate to short (PS-Q1 = 3.7 ± 0.98), with minimal post-extraction complaints (PS-Q2= 4.3 ± 0.9). Most patients would undergo the treatment again, when necessary (PS-Q3 = 4.7 ± 0.7), and nearly all would recommend FIIPP to others (PS-Q4 = 4.8 ± 0.4). The high total PS score at T2 was mainly caused by the high scores of Q5–8 concerning the esthetics. After three years, the color of the gingiva (Q5 = 4.5 ± 0.8), the shape and location of the gingival margin (Q6 = 4.4 ± 0.8), and the implant crown (Q8 = 4.8 ± 0.4) were judged as beautiful and the look of the gums as “natural” (Q7 = 4.4 ± 0.8). No correlation was observed between PS and PES (Pearson correlation 0.099; *p* = 0.372), and the same was true for PS and WES (Pearson correlation 0.066; *p* = 0.554).

### 3.6. Soft-Tissue Esthetic Outcome

The inter-examiner reliability was tested after calibration on the first 112 PES scores at T0 using a paired *t*-test (examiner 1 = 9.75 ± 2.84; examiner 2 = 9.63 ± 2.89; *p* = 0.91). Figure 2 demonstrates that the pink esthetics improved stepwise, from a preoperative (T0) PES of 9.9 (±2.5) to a 3-year postoperative (T4) PES of 12.2 (±1.6), and a preoperative modPES of 6.7 (±1.8) to a T4-3-year postoperative modPES of 8.5 (±1.4). Independent of which score was used, soft-tissue esthetics improved significantly in time (effect 1.931/1.013).

### 3.7. Tooth-Related Esthetic Outcome

The inter-examiner reliability was tested using a paired *t*-test on the first 80 preoperative (T0) WES scores of both examiners (examiner 1 = 4.5 ± 2.87; examiner 2 = 4.5 ± 2.83; *p* = 1). The mean preoperative WES was 4.4, and increased to 4.8 directly after (T1); an improvement of 3.437 points was found after replacement of the temporary for the final crown (T2), which remained stable after one and three years (T3: WES = 8.2; T4: WES = 8.2) (Figure 2).

### 3.8. Classification of Esthetic Outcomes

Classification of PES is shown in Table 3; after three years, in 72% of the cases, the outcome was labelled as “excellent” (PES = 12–14), and “inadequate” (PES < 8) in 2%. The overall esthetic scores (PES and WES) at baseline and after three years are shown in Table 3; a “good to excellent” and an “inadequate” overall esthetic outcome was achieved in 75% and 6% of patients.

Light photographs of patients with an “inadequate” overall esthetic outcome (*n* = 5; PES < 8 and/or WES < 6) after three years are depicted in Figure 3.

### 3.9. Multilevel Regression Analysis (MRA)

The outcomes of factors that may be associated with the esthetic outcome (PES/modPES/WES) are listed in Table 4. The hard- and soft-tissue outcomes were not associated with smoking and/or the size of post-extraction bone defects. The emergence profile had the highest impact on the soft-tissue esthetics; concave emergence profiles scored significantly higher (1.3 PES points; *p* = 0.00) than convex emergence profiles.

Preoperative esthetics: no effect on PES, and a very low effect on modPES and WES, with higher pre-extraction scores associated with higher scores three years later;

Smoking: no effect;

Post-extraction bone defects (mm): no effect;

Implant location: central incisor locations had significantly higher soft-tissue esthetic results compared to lateral incisors but was no effect on WES;

Gap size (mm): larger buccal gaps showed significantly lower PES and modPES, and note that the average gap size was 2.7 ± 0.9 mm (range 1.3–6.4 mm);

Final crown (cemented versus screw retained): cemented crowns resulted in a significantly higher WES compared to screw-retained crowns;

Emergence profile (convex versus concave): significantly lower pink and white esthetics were achieved with a convex emergence profile;

Complications: only the modPES showed significantly lower scores;

Time: postoperatively, WES improved stepwise from T0 to T4, with an improvement of 3.437 points after the temporary crown (T1) was replaced by a final crown (T2) (*p* < 0.001), and thereafter remained stable; the PES/modPES scores showed a different dynamic, where there was a marked improvement immediately after FIIPP (T1), followed by a slow increase until T4 and this can be concluded from the combined linear and quadratic time term in the MRA model.

## 4. Discussion

After three years, patient satisfaction and the objective esthetic scores (PES/modPES/WES) remained high compared to the one-year outcomes, supporting the hypothesis. The overall patient satisfaction did not correlate with the achieved esthetic outcome. Based on the multiple esthetic parameters evaluated, it may be concluded that patients are more satisfied than the objective esthetic measurements predict.

The size of post-extraction buccal bone defects could not be associated with the soft-tissue esthetic outcome. In contrast, implant locations do matter: both PES and modPES scores were significantly higher for the location ‘central incisor’ than the ‘lateral incisor’. As lateral incisors have narrower sockets, less space is available to create a gap of at least 2 mm wide in front of an implant, even when a smaller implant diameter of 3.0 mm is selected.

WES improved significantly with cemented final crowns (vs. screw retained) and when a concave emergence profile (vs. convex) was provided. An explanation for the lower esthetic outcome with screw-retained crowns may be that, in general, compared to cemented crowns with a custom-made abutment underneath, less dimension is available for the ceramics at the buccal due to the implant’s direction and the dimension/position of the screw access channel, leading to less translucency, color mismatch and/or over contouring. The negative effect of convex emergence profiles on the esthetic outcome of the crown and on the soft-tissue esthetics may be explained by the appearance of too “bulky” crowns and excessive pressure on the soft and hard tissues during placement of the final crown, respectively.

The negative effect of larger gap sizes may be interpreted as follows: gaps can also be too wide, as wider gaps form a higher risk for the bone substitute to be washed out. Wash out may be prevented by using bone substitute blocks instead of loose particles.

In the first year post extraction, both WES and PES improved compared to before surgery. After three years, “inadequate overall esthetics” diminished to 6% from the baseline 67%. These results compare very well to other studies reporting higher esthetic failures ranging between 21% and 47% with single implant treatments [20,25,26,29,37,45,46]. Five patients were considered esthetic failures in the present study. One of two patients scoring “inadequate PES” encountered the following surgical and restorative complications: loss of the buccal crest during extraction, rupturing the soft tissues, loosening of the temporary crown and a buccal fistula which compromised the bone graft. The second had a pre-existing buccal recession, which was pushed even more apically by an over contoured final crown. Three patients scored “inadequate WES”: a 21-year-old female patient’s neighboring teeth became ‘longer’ compared to the implant crown due to maxillary vertical growth, while the remaining two patients had asymmetry and thus an esthetically inadequate situation already at baseline. Unexpectedly, all patients showing an inadequate PES and/or WES were satisfied.

Comparable research is listed in Table 5. Regardless of the timing of implant placement, studies recommending a more buccal implant position report lower and/or unstable soft-tissue outcomes [17,19,20,25,26,28,29,31,35,37,46,47]. Meanwhile, in studies with a more palatal implant position, higher esthetic outcomes are observed [30,36]. The reason for this may be that more space is available for vascularization of the socket preservation/graft in front of the implants, supporting and stabilizing the overlying soft tissues. Meta-analysis, based on one randomized clinical trial (RCT), demonstrated 54% less horizontal buccal bone resorption following IIP with socket graft (SG) when compared to IIP alone [48].

In addition to a more palatal implant seat position, a flapless approach and/or immediate restoration may also contribute to higher soft-tissue esthetics. A prospective case series on IIP [26], raising a flap and no provisionalization showed poorer soft-tissue outcomes, PES = 10.48 ± 2.47, and a higher percentage (16%) of “inadequate PES” [26]. There is some evidence that a flapless approach results in more buccal bone preservation [49], and immediate provisionalization may contribute to mid-buccal soft-tissue stability at immediate implants [50]. Nevertheless, high-quality RCTs are necessary to confirm the clinical relevance of these findings.

Some authors suggested that applying a connective tissue graft (CTG) is a good choice when a high risk of mid-facial recession is identified [51]. It is also stated, that CTG can be avoided when the risk of recession is low [52]. However, risk assessments remain difficult, especially when statements are based on studies with a more buccal implant position and with a high variability in materials and treatment methods. In the current study with implant positions maximal to the palatal, three years post extraction, a ST recession of ≥2 mm was still present (compared to 16 at baseline) in six patients, and a severe alveolar process deficiency was found in four patients (see Table 6). In those cases, an additional CTG could be considered. Nevertheless, none of these patients were unsatisfied with their treatment outcome: patients who scored a ‘0′ for the PES-3 after 3 years had an average PS of 9.2 (range 7.1–10), and patients who scored a ‘0′ for PES-5 after 3 years had a mean PS of 8.4 (range 6.9–9.6).

Considerably lower soft-tissue esthetic outcomes were reported for DIP compared to EIP, IIP, and FIIPP. Recent prospective clinical research reported a one-year modPES of 6.9, which declined (non-significantly) to 6.6 at five years. In 74% of cases, a modPES ≥6 was found, which implies that a poor soft-tissue outcome was observed in the remaining 26% [20].

The esthetic outcome cannot be associated with the timing of implant placement [30,35] but the chosen protocol can. In a randomized clinical trial, a significantly lower pink esthetic outcome for IIP was reported compared to DIP [25]. In both IIP and DIP, primary flap closure was performed, resulting in 26% wound healing complications in IIP, compared to 5% in DIP. Obviously, when performing IIP, primary wound closure should not be pursued.

When a failing tooth is still in situ, the choice between FIIPP and EIP is crucial. The presented study, including intact, as well as defect sockets, showed similar, or even higher, PES compared to EIP studies [14,15,17]. In addition to achieving high esthetics with this protocol, FIIPP is the most attractive treatment from the patients’ perspective, as it is less invasive than conventional protocols, patients are immediately provided with a fixed solution, and only four visits are needed.

If immediate implants are contraindicated, for instance due to dental or general health conditions, DIP after ridge preservation (RP) seems to be a good alternative [30]. Nowadays, tooth extraction without performing at least a RP may be questionable. Already in 2012, consensus was reached that RP is indicated when a tooth is considered lost and future implant treatment likely to preserve the alveolar volume. This is of course especially true for the esthetic zone, namely the upper front teeth.

One weakness of the described FIIPP procedure is that the implants are placed deep below the reconstructed buccal crest, making them susceptible to infection during impression taking or placement of the final abutment. To reduce biological complications, the authors suggest inserting the final abutment during the surgical stage.

A limitation of this prospective clinical trial is the absence of a control group, such as an EIP or a DIP approach. However, the goal was to validate the presented FIIPP method first. Another limitation of this study is that the indices used to assess esthetics are only subjective methods. Linear or volumetric soft-tissue changes provide a more objective assessment on soft-tissue changes over time. Additionally, it should be noted that the longevity of the esthetic outcome depends not only on the surgical and restorative procedure used, but also on good oral hygiene and correct occlusion/articulation.

In conclusion, the 3-year postoperative esthetic outcomes remained high compared to the 1-year results. Buccal bone defects form no contraindication for FIIPP. However, creating gaps that are too wide and convex emergence profiles should be avoided.

## Figures and Tables

**Figure 1 jcm-12-02625-f001:**
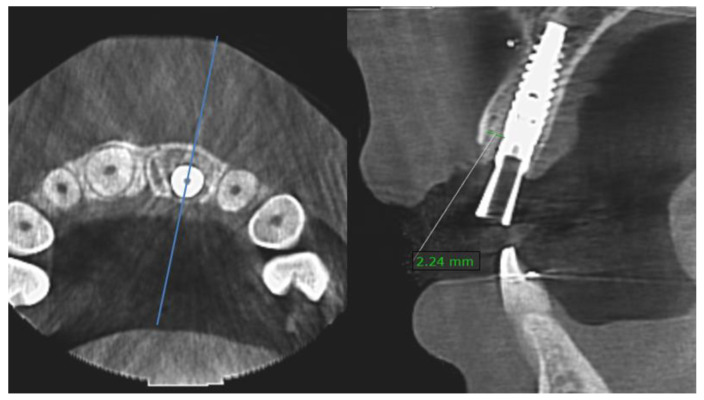
Cross-sectional measurement of the mid-buccal (blue line) gap size using CBCT.

**Figure 2 jcm-12-02625-f002:**
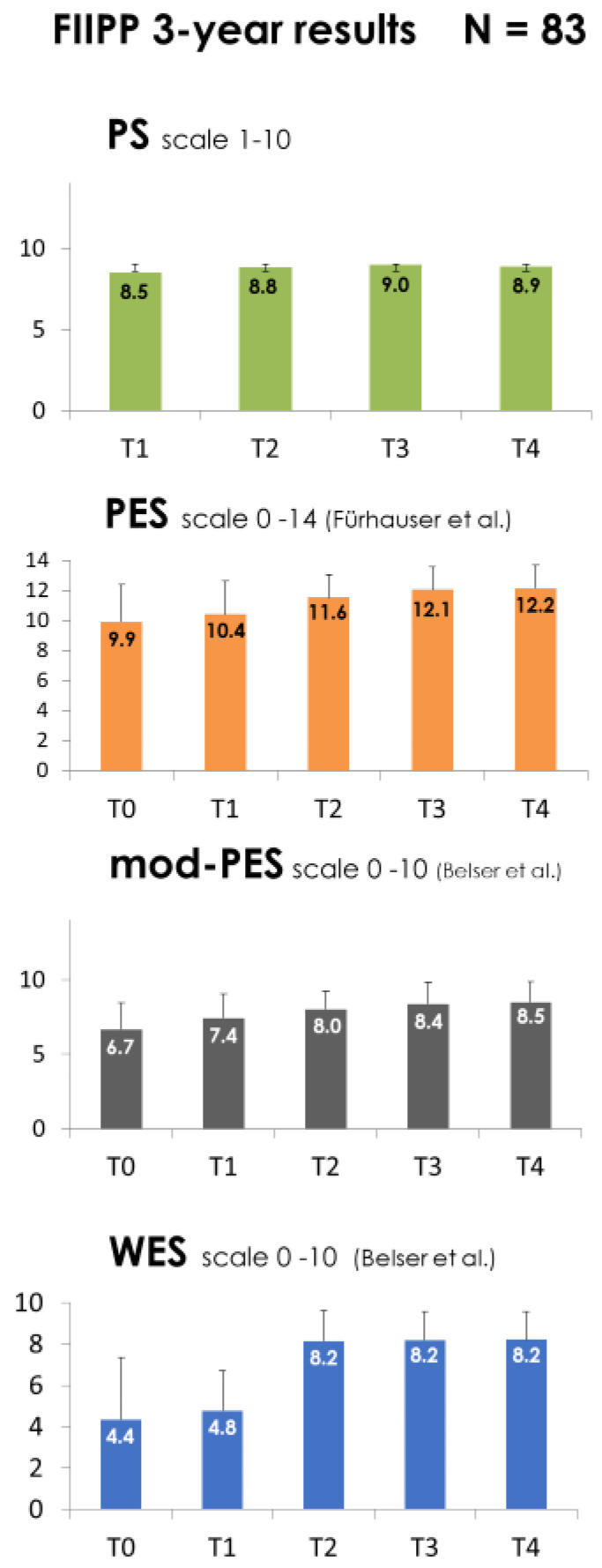
Mean and SD of patient satisfaction (PS), pink esthetic score (PES), modified pink esthetic score (modPES), and white esthetic outcome (WES) at different time points; preoperative (T0), 7–14 days after (T1), directly after crown placement (T2), 1 year postoperative (T3), and 3 years postoerative (T4) [15,16].

**Figure 3 jcm-12-02625-f003:**
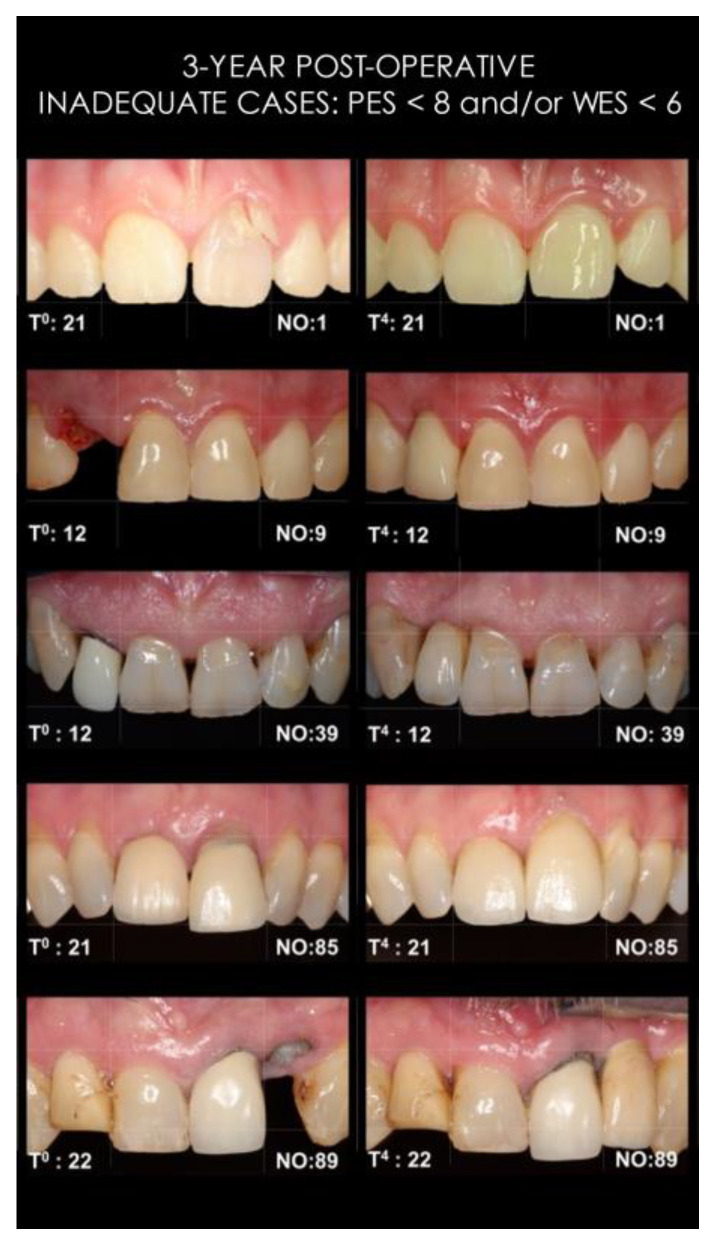
“Inadequate” esthetic outcomes at T4; no. 1 and 9 (WES = 5), no. 89 (PES = 3), no. 39 (PES = 7), and no. 85 (PES = 6). All patients were satisfied with the esthetic outcome (PS = 6.7, 9.8, 8.9, 9.6, and 9.3 for no. 1 to no. 89, respectively).

**Table 1 jcm-12-02625-t001:** Patient Satisfaction Questionnaire.

Patient Satisfaction Questionnaire		
Q1	What was your experience concerning duration of surgery?	1–5: long–short
Q2	What was your experience about post-surgery complaints?	1–5: severe–limited
Q3	Would you undergo this treatment again in a similar situation?	1–5: no–certainly
Q4	Would you recommend this treatment to others?	1–5: no–certainly
Q5	What is your opinion about color/gums around implant?	1–5: ugly–beautiful
Q6	What is your opinion about shape/location of gums margin?	1–5: ugly–beautiful
Q7	Do the gums look natural compared to the natural teeth?	1–5: no–certainly
Q8	What is your opinion about the form/color of the implant crown?	1–5: ugly–beautiful
Q9_1–5_	What is your total score considering the treatment and result	1–5: poor–excellent
Q9_1–10_	Total score (Q9: range 1–5) transformed to a scale of 1–10.	1–10: poor–excellent

**Table 2 jcm-12-02625-t002:** Size of post-extraction mid-buccal bone defects (MBBDs) measured in mm related to their 3-year soft-tissue esthetic outcome (T4-PES).

T4-PES	6	7	8	9	10	11	12	13	14	n
MBBD (mm)										
0				2		5	8	8	6	29
1					1	5	3	6	3	18
2	1		1	1		2	1	7	2	15
3							2	1	1	4
4						1		4	1	6
5										
6							1	3		4
7		1					1	1		3
8										
9						1		1		2
10										
11							1			1
12										
13					1					1
n	1	1	1	3	2	14	17	31	13	83

**Table 3 jcm-12-02625-t003:** (a) Ranking of “pink esthetic outcomes” using the original PES. (b) “Overall esthetic scores” using a combination of original PES and WES.

a. Ranking of Pink Esthetic Outcomes: PES Score (Scale 0–14) N = 83				
	Excellent PES 12–14	Acceptable PES 8–11		Inadequate PES 0–7
**0 year (T0)**	33%	48%		19%
**1 year (T3)**	69%	30%		1%
**3 year (T4)**	72%	26%		2%
**b. Overall Esthetic Outcome: Original PES Score and WES N = 83**				
	Excellent PES ≥ 12/WES≥9	Good PES ≥ 10/WES ≥ 8	Acceptable PES ≥ 8/WES ≥ 6	Inadequate PES < 8/WES < 6
**0 year (T0)**	8%	5%	20%	67%
**1 year (T3)**	37%	37%	20%	6%
**3 year (T4)**	40%	35%	19%	6%

**Table 4 jcm-12-02625-t004:** Factors associated with the esthetic results after FIIPP (* means *p* ≤ 0.01).

PES			
	**Effect**	**95% CI**	** *p* **
Intercept	11.79	[9.82…13.78]	0.00 *
Preoperative esthetics (T0-PES)	0.06	[−0.03…0.15]	0.20
Smoking	−0.12	[−0.74…0.50]	0.73
Post-extraction bone defect (mm)	0.04	[−0.06…0.13]	0.44
Implant location (lateral vs. central)	−0.85	[−1.30…−0.40]	0.00 *
Gap size (mm)	−0.41	[−0.67…−0.15]	0.01 *
Crown (cemented vs. screwed)	0.19	[−0.31…0.68]	0.48
Emergence profile (convex vs. concave)	−1.26	[−1.76…−0.76]	0.00 *
Complication	−0.59	[−1.20…0.02]	0.07
Time	1.93	[1.18…2.68]	0.00 *
Time^2^	−0.27	[−0.42…−0.12]	0.00 *
** modPES **			
	**Effect**	**95% CI**	** *p* **
Intercept	8.30	[6.68…9.94]	0.00 *
Preoperative esthetics (T0-modPES)	0.11	[0.01…0.20]	0.05
Smoking	0.02	[−0.50…0.54]	0.94
Post-extraction bone defect (mm)	0.06	[−0.02…0.13]	0.16
Implant location (lateral vs. central)	−0.57	[−0.94…−0.20]	0.01 *
Gap size (mm)	−0.29	[−0.51…−0.08]	0.01 *
Crown (cemented vs. screwed)	0.11	[−0.30…0.51]	0.61
Emergence profile (convex vs. concave)	−0.82	[−1.23…−0.41]	0.00 *
Complication	−0.66	[−1.16…−0.16]	0.02
Time	1.01	[0.36…1.66]	0.00 *
Time^2^	−0.13	[−0.26…−0.00]	0.05
** WES **			
	**Effect**	**95% CI**	** *p* **
Intercept	4.78	[3.20…6.36]	0.00 *
Preoperative esthetics (T0-WES)	0.09	[0.02…0.16]	0.02
Smoking	−0.28	[−0.85…0.29]	0.36
Post-extraction bone defect (mm)	0.01	[−0.08…0.09]	0.83
Implant location (lateral vs. central)	−0.27	[−0.69…0.16]	0.24
Gap size (mm)	0.01	[−0.24…0.25]	0.97
Final crown (cemented vs. screwed)	0.57	[0.12…1.01]	0.02
Emergence profile (convex vs. concave)	−0.68	[−1.14…−0.23]	0.01 *
Complication	−0.19	[−0.74…0.37]	0.53
Time is >1	3.44	[3.10…3.77]	0.00 *

**Table 5 jcm-12-02625-t005:** (a) Prospective esthetic outcomes of four IIP studies, using Bio-Oss™ as a filler of the socket. Except of the study of Cosyn et al. [26], a flapless protocol was followed combined with immediate loading. (b) Prospective esthetic outcomes on single maxillary tooth replacements performing various protocols: IIP, EIP, and DIP. contrast of to EIP/DIP protocols, the IIP protocols followed a flapless approach, except for Tonetti et al. [25], who performed primary closure of the socket in IIP. Except for Raes et al. [46,47], bone (substitute) and a membrane were used in all studies.

a. Comparable IIP Studies						
Study	N	Socket	Implant	Implant Position	Esthetic Outcome	Other
Cosyn et al. [26]	25	Intact	Nobel Replace	H: 1 mm P V: 1 mm A	3-year results: PES = 10.5/WES = 8.2	PES: 16% inadequate
Cosyn et al. [29]	17	Intact	Nobel Active	H: 1 mm P V: 1 mm A	1-year: PES = 12.2 5-year: PES = 11.2	MB recession: 47%
Seyssens et al. [37]	18	Intact	Nobel Active	H: 1 mm P V: 1 mm A	10-year results: PES = 10.6	MB recession: 33% advanced
Slagter et al. [24]	20	Intact	Nobel Active	H: 1 mm P V: ≥3 mm A	5-year results: PES = 7.8/WES = 7.5	MB recession: mean 1.44 mm
** b. Comparable Studies with Different Treatment Timing **						
** Study **	** N + Timing **			** Implant Position **	** Healing Protocol **	** Esthetic Outcome **
Raes et al. [46]	15 IIP 23 DIP			H: 1 mm P V: 1 mm A	temp crown	1-year results: IIP: PES = 10.3 DIP: PES = 9.7
Buser et al. [17]	20 EIP			H: 1 mm P V: 1 mm A	submerged	3-year results: modPES = 8.1 WES = 8.7
Buser et al. [19]	41 EIP			H: 1 mm P V: 1 mm A	submerged	5-year results: modPES = 7.8/WES = 7 9-year results: modPES = 7.5/WES = 6.9
Felice et al. [30]	25 IIP 25 RP and DIP			H: 1.5 mm P V: 4–5 mm MB	temp crown	1-year results: IIP: PES = 12.8/ DIP: PES = 12.2
Tonetti et al. [25]	58 IIP 57 DIP			not reported	submerged IIP: 62% DIP: 82%	1-year IIP: modPES ≈ 7 42% inadequate 1-year DIP: modPES ≈ 8 19% inadequate
Arora and Ivanovski [35]	15 IIP 15 EIP			H: 1 mm P V: 1 mm	healing abutment	1-year post crown: IIP: PES = 9.4 EIP: PES = 9.3
Raes et al. [47]	11 IIP 18 DIP			H: 1 mm P V: 1 mm	temp crown	>8-year results: IIP: PES = 10.4 DIP: PES = 9.2
Meijndert et al. [20]	50 DIP			H: 1 mm PV: 1 mm	submerged	1- and 5-year results: modPES = 6.9 and 6.6 WES = 7.5 and 7.8

IIP = immediate implant placement; DIP = delayed implant placement; EIP = early implant placement; RP = ridge preservation; PES = pink esthetic score; modPES = modified pink esthetic score; WES = white esthetic score; H = horizontal; V = vertical; P = palatal; A = apical of cement–enamel junction; MB = mesio-buccal.

**Table 6 jcm-12-02625-t006:** Shows the frequency distribution, mean and standard deviation of soft-tissue levels and alveolar process deficiencies of the current research at baseline (T0) and after three years (T4).

	T0					T4				
	Frequency			Mean	SD	Frequency			Mean	SD
Score	0	1	2			0	1	2		
PES-3: soft-tissue level	16	29	38	1.27	0.76	6	20	57	1.61	0.62
PES-5: alveolar process contour	2	16	65	1.76	0.48	4	24	55	1.61	0.58
Total	83					83				

## Data Availability

The data presented in this study are available on reasonable request from the corresponding author. The data are not publicly available due to privacy and ethical reasons.

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
