# Peer review of "Factors Associated with Esthetic Outcomes of Flapless Immediate Placed and Loaded Implants in the Maxillary Incisor Region—Three-Year Results of a Prospective Case Series"

_jcm, 2023, doi:10.3390/jcm12072625_

Round 1

Reviewer 1 Report

The paper is very well written and provides extensive case support for the aesthetic results of performing immediate implantation and loading without flap in cases of single tooth loss in the anterior aesthetic zone. There are few problems, which must be solved before it is considered for publication. If the following problems are well-addressed, this reviewer believes that the essential contribution of this paper are important for the immediate implant placement and loading of the anterior aesthetic zone.

1. The meaning of some statements is confusing, for example L57, and L394.

2. It would be better if there is a figure to show the measurement of the gap size.

3. It is suggested to label the cases of P<0.01 in Table 4 to make the results more obvious.

4. Your manuscript needs careful editing and particular attention to English grammar, spelling, and sentence structure.

Author Response

Response to Reviewer 1 Comments

Dear Reviewer,

Thank you for your valuable comments and positive feedback.

In the lines below you find our response in red. We've been following up on all your suggestions. Please note that the lines don’t correspond with the first version since the manuscript underwent careful editing in which particular attention was paid to English grammar, spelling, and sentence structure.

Reviewer 2 Report

This is an interesting 3-year follow-up on 100 patients who have been treated with an immediate implant in the esthetic zone. The paper is well-written, yet could be improved on the basis of the following comments. Please respond.

1/ Please rephrase the title as follows: “Factors associated with esthetic outcomes of flapless immediately placed - and loaded implants in the maxillary incisor region. Three-year result of a prospective case series.” A case series does not allow to establish risk factors that affect an outcome. Here, only associations can be made and therefore, the word “affect” should be avoided. Also pay attention to this wording in the abstract and text please.

2/ In the selection criteria it is mentioned that only buccal bone defects <= 5 mm were allowed (line 97). However, in the results section it is written that in nine patients post-extraction buccal bone defects surpassed 5 mm (line 202, 203). This is contradictory.

3/ How did the authors evaluate the emergence profile? As this is hidden below the soft tissue margin, it is not possible to evaluate it clinically. Please clarify in the materials and methods section.

4/ The authors found inferior esthetics for screw-retained crowns, yet these were oversized due to an angulated screw channel abutment. The morphology of the abutment is a confounder in this context. Therefore, it may not be correct to state that screw retention related to an inferior esthetic outcome. The authors should assess multicollinearity since a correlation between the mode of retention and the emergence profile is suspected. Only if there is no multicollinearity, both factors can be considered in the regression model. Please revise the statistical analysis and adapt the results/conclusions accordingly.

5/ The authors found inferior esthetics when the gap size increased. This does not seem in line with recent findings of Levine et al. (2022) showing thicker buccal bone with increasing gap size. Please discuss.

6/ The indices used by the authors are only one method to assess esthetics. The problem is that these are composite indices and their assessment remains subjective. Linear or volumetric soft tissue changes provide a more objective assessment on soft tissue changes over time. The authors should acknowledge this weakness in the discussion.

7/ “Blinded examiners” assessed the esthetic outcome (line 153). This term applies to an RCT, yet this concerns a case series as all patients received the same treatment. Please remove “blinded”.

8/ Recent systematic reviews on immediate implant placement are missing. These highlight the relevance of socket grafting and immediate provisionalization in terms of soft tissue preservation, which are two concepts that have been applied in this case series. Please update the reference list.

9/ The authors should elaborate on the need for soft tissue grafting in the light of recent systematic reviews. “Alveolar process” is one of the 7 criteria of the PES and provides insight in this. Please share the results on alveolar process deficiency in detail and discuss the need for soft tissue grafting on the basis of the amount of cases demonstrating such deficiency.

10/ I really appreciated figure 2 with all the failure cases. This is a nice and transparent way of reporting.

Author Response

Response to Reviewer 2 Comments

Dear Reviewer,

Thank you for your valuable comments and positive feedback. Abusively, I send you the wrong response (of reviewer 3), please consider this one as not send.

Hereby you find our response in red. We've been following up on all your suggestions. Please note that the lines don’t correspond with the first version since the manuscript underwent careful editing in which particular attention was paid to English grammar, spelling and sentence structure.

1/ Please rephrase the title as follows: “Factors associated with esthetic outcomes of flapless immediately placed - and loaded implants in the maxillary incisor region. Three-year result of a prospective case series.” A case series does not allow to establish risk factors that affect an outcome. Here, only associations can be made and therefore, the word “affect” should be avoided. Also pay attention to this wording in the abstract and text please. Made changes throughout the manuscript in confirmation with above mentioned see title, abstract and results( lines 310-347) and discussion.

2/ In the selection criteria it is mentioned that only buccal bone defects <= 5 mm were allowed (line 97). However, in the results section it is written that in nine patients post-extraction buccal bone defects surpassed 5 mm (line 202, 203). This is contradictory. Inclusion of bone defects (≤ 5 mm) was based on pre-operative CBCT scans with the cement-enamel junction as reference.

Post-extraction the buccal crest was evaluated by measuring "from the top of the mid-buccal gingiva to the buccal crest" by use of a periodontal probe. The actual bone defects were defined as: the distance "from the top of the mid-buccal gingiva to the buccal crest" minus 3 mms. For clarification, textual corrections are made accordingly.

3/ How did the authors evaluate the emergence profile? As this is hidden below the soft tissue margin, it is not possible to evaluate it clinically. Please clarify in the materials and methods section. See line 157-158: "Emergence profile of the final crown was evaluated on CBCT and defined "convex" when a straight or round profile was present, or "concave" when the profile was slender".

4/ The authors found inferior esthetics for screw-retained crowns, yet these were oversized due to an angulated screw channel abutment. The morphology of the abutment is a confounder in this context. Therefore, it may not be correct to state that screw retention related to an inferior esthetic outcome. The authors should assess multicollinearity since a correlation between the mode of retention and the emergence profile is suspected. Only if there is no multicollinearity, both factors can be considered in the regression model. Please revise the statistical analysis and adapt the results/conclusions accordingly. I totally agree at this point, however within the short timespan of only 5 days (including a weekend) for revision, it is impossible to revise the statistical analysis and adapt the results. Instead, I pointed this out in the discussion and changed the clinical implications and conclusions. I hope you can agree on this.

Line 369-371: "The former may be explained by the wider diameter of the angulated screw channel (ASC) abutments used to fabricate the screw-retained crowns. “Bulky” crowns arise, from an abutment-diameter which is wider than the outer contour of the implant seat."

5/ The authors found inferior esthetics when the gap size increased. This does not seem in line with recent findings of Levine et al. (2022) showing thicker buccal bone with increasing gap size. Please discuss. Levine et al devided the population in 2 groups (which is statistically not so strong): Wide gaps of > 2 mm ( at implant shoulder level  m=1.7 ± 1, range 1.3 - 2.4) or narrow gaps of ≤ 2 mm (at implant shoulder level m=0.5 ± 0.8, range 0-1.1).The so called wide gaps, were not so wide as in current study, and the narrow gaps are to small to expect a proper revascularization. So it is evident that the s called wide gaps performed better.

The gaps in current study are much wider: Mean gap size was 2.7 ± 0.9 mm (range 1.3 – 6.4 mm). Addtionally, in this study a MRA was performed, no classification

6/ The indices used by the authors are only one method to assess esthetics. The problem is that these are composite indices and their assessment remains subjective. Linear or volumetric soft tissue changes provide a more objective assessment on soft tissue changes over time. The authors should acknowledge this weakness in the discussion. I agree, mentioned this in the discussion lines 459-461: " Another limitation of this study is that the indices used are only subjective methods were used to assess esthetics. Linear or volumetric soft tissue changes provide a more objective assessment on soft tissue changes over time."

7/ “Blinded examiners” assessed the esthetic outcome (line 153). This term applies to an RCT, yet this concerns a case series as all patients received the same treatment. Please remove “blinded”. Line 170: changed into "independent examiners"

8/ Recent systematic reviews on immediate implant placement are missing. These highlight the relevance of socket grafting and immediate provisionalization in terms of soft tissue preservation, which are two concepts that have been applied in this case series. Please update the reference list.

See line 444-446: "The advantage of the presented FIIPP method, combining socket preservation by use of a bone substitute and provisionalization to support the soft tissues, is confirmed in a recent randomized clinical trial [50].

  1. Girlanda, F.F, Hsu Shao Feng, H.S, Corrêa, M.G, Casati, M.Z., Peres Pimentel, S., Vieira Ribeiro, F., Ribeiro Cirano, F. Depro-teinized bovine bone derived with collagen improves soft and bone tissue outcomes in flapless immediate implant approach and immediate provisionalization: a randomized clinical trial. Clin Oral Investig 2019;23(10):3885-3893. doi: 10.1007/s00784-019-02819-x 

9/ The authors should elaborate on the need for soft tissue grafting in the light of recent systematic reviews. “Alveolar process” is one of the 7 criteria of the PES and provides insight in this. Please share the results on alveolar process deficiency in detail and discuss the need for soft tissue grafting on the basis of the amount of cases demonstrating such deficiency.

See line 436-438: "Moreover, it should be considered that the use of a connective tissue graft could also play a role. It seems to be a good choice when there is a high risk of mid-facial recession [48], whereas it seems that it can be avoided when that risk is low [49]."

  1. Seyssens, L., De Lat, L., Cosyn, J. Immediate implant placement with or without connective tissue graft: A systematic review and meta-analysis. J Clin Periodontol. 2021;48(2):284-301. doi:10.1111/jcpe.13397
  2. Ferrantino, L., Camurati, A., Gambino, P., Marzolo, M., Trisciuoglio, D., Santoro, G., Farina, V., Fontana, F., Asa'ad, F., Simion,
  3. Aesthetic outcomes of non-functional immediately restored single post-extraction implants with and without connective tissue graft: A multicentre randomized controlled trial. Clin Oral Implants Res. 2021;32(6):684-694. doi: 10.1111/clr.13733

10/ I really appreciated figure 2 with all the failure cases. This is a nice and transparent way of reporting. Thank you ?

Reviewer 3 Report

Dear Author,

I appreciate the effort you have put into your manuscript, which appears to me a very good one. I only have some minor issue that might be useful to further improve the quality of your research:

INTRODUCTION

1. line 78 - DIP should be in parentheses

M&M

2. line 98 - It was just bruxism in the previous publication, why it is now “extreme”?

3. line 128 - Please specify that it was a non functional provisional crown

4. line 136 - G-B appears only here in the whole document, consider its use.

5. line 162 - It seems that “≥” is missing

6. line 182 - Please add a definition of the variable “time” you used in the MRA analysis. It seems you used it as a continuous variable. Is it the number of days (or weeks or months) from T0?

RESULTS

7. line 215 - Might be used the above defined “G-B” to be more clear

8. line 217 - It seems to me that the table 2 showed 11 buccal bone defect larger than 5 mm. Please clarify

9. line 253 and line 261 - PES and WES are numerical variable by definition, but they do not meet all the criteria to be automatically considered as a continuous, normally distributed variable. Could you please explain: 1) On which basis did you considered PES and WES as continuous, normally distributed variables? it has been performed any test to confirm the assumption? 2) Assuming that PES and WES can be considered continuous and normally distributed, why the inter-examiner reliability was tested with a paired t-test rather than an Intraclass Correlation Coefficent?

10. line 294 - "Implant location" should be bold

DISCUSSION

11. Table 5b - "RP" has not been defined before.

12. line 379 - Regarding the chosen protocol, i totally agree with your considerations on flap closure and ridge preservation. Moreover, it should be the case to consider that the use of a connective tissue graft could also play a role. It seems to be a good choice in cases of risk of mid-facial recession (Seyssens L, et al. 2021) whereas it seems that it can be avoided when that risk is low (Ferrantino L. et al. 2021).

13. line 390 and line 391 - RP should be defined when you use it for the first time in the manuscript

14. line 409 - On what basis the gap mean value has been chosen as the threshold between positive and negative effects on the FIIPP healing process?

Best regards 

Author Response

Dear Reviewer,

Thank you for your valuable comments. As an attachment, you find the corresponding answers.

With best regards, Edith Groenendijk

Round 2

Reviewer 2 Report

Overall, the authors improved their paper, but some responses were unsatisfactory. Only when the authors adequately respond to earlier comments, the paper can be accepted for publication. Please respond to the following: 

1/ Remark n° 4: The authors found inferior esthetics for screw-retained crowns, yet these were oversized due to an angulated screw channel abutment. The morphology of the abutment is a confounder in this context. Therefore, it may not be correct to state that screw retention related to an inferior esthetic outcome. The authors should assess multicollinearity since a correlation between the mode of retention and the emergence profile is suspected. Only if there is no multicollinearity, both factors can be considered in the regression model. Please revise the statistical analysis and adapt the results/conclusions accordingly. 

The authors agreed but basically did nothing with it in the revised paper. Having not enough time is not an excuse to revise the analysis. If the journal only provides 5 days, you should ask for additional time. Please revise the statistical analysis and adapt the results/conclusions accordingly. 

2/ Remark n° 8: Recent systematic reviews on immediate implant placement are missing. These highlight the relevance of socket grafting and immediate provisionalization in terms of soft tissue preservation, which are two concepts that have been applied in this case series. Please update the reference list. 

The authors added the Girlanda paper as a response to this comment. Please add recent systematic reviews on socket grafting as well as on soft tissue grafting instead, as they combine the results of multiple RCTs. Therefore, their level of evidence is higher than one RCT out of many. 

3/ Remark n°9: The authors should elaborate on the need for soft tissue grafting in the light of recent systematic reviews. “Alveolar process” is one of the 7 criteria of the PES and provides insight in this. Please share the results on alveolar process deficiency in detail and discuss the need for soft tissue grafting on the basis of the amount of cases demonstrating such deficiency. 

The authors added a sentence in the discussion on the need for soft tissue grafting, yet the authors were asked to share the detailed results on “alveolar process”. Please do so and comment on your own results in light of existing evidence.

Author Response

Response to Reviewer 2 Comments

Dear reviewer,

My apologies that the answers below were responded insufficiently; I thank you for your quick response on this.

As an attachment, you find the revision

Best Regards, Edith Groenendijk
